# Understanding the Synergy of NKp46 and Co-Activating Signals in Various NK Cell Subpopulations: Paving the Way for More Successful NK-Cell-Based Immunotherapy

**DOI:** 10.3390/cells9030753

**Published:** 2020-03-19

**Authors:** Loris Zamai, Genny Del Zotto, Flavia Buccella, Sara Gabrielli, Barbara Canonico, Marco Artico, Claudio Ortolani, Stefano Papa

**Affiliations:** 1Department of Biomolecular Sciences, University of Urbino “Carlo Bo”, 61032 Urbino, Italy; 2INFN-Gran Sasso National Laboratory, Assergi, 67100 L’Aquila, Italy; 3Area Aggregazione Servizi e Laboratori Diagnostici, IRCCS Istituto Giannina Gaslini, 16147 Genoa, Italy; 4Department of Sensory Organs, Sapienza University of Rome, 00161 Rome, Italy

**Keywords:** NK cell biology, NK cell subsets, NK cell activating receptors, cell adhesion molecules, granule polarization, degranulation, cytotoxicity assay, trogocytosis, long-lived memory-like NK cells

## Abstract

The NK cell population is characterized by distinct NK cell subsets that respond differently to the various activating stimuli. For this reason, the determination of the optimal cytotoxic activation of the different NK cell subsets can be a crucial aspect to be exploited to counter cancer cells in oncologic patients. To evaluate how the triggering of different combination of activating receptors can affect the cytotoxic responses of different NK cell subsets, we developed a microbead-based degranulation assay. By using this new assay, we were able to detect CD107a^+^ degranulating NK cells even within the less cytotoxic subsets (i.e., resting CD56^bright^ and unlicensed CD56^dim^ NK cells), thus demonstrating its high sensitivity. Interestingly, signals delivered by the co-engagement of NKp46 with 2B4, but not with CD2 or DNAM-1, strongly cooperate to enhance degranulation on both licensed and unlicensed CD56^dim^ NK cells. Of note, 2B4 is known to bind CD48 hematopoietic antigen, therefore this observation may provide the rationale why CD56^dim^ subset expansion correlates with successful hematopoietic stem cell transplantation mediated by alloreactive NK cells against host T, DC and leukemic cells, while sparing host non-hematopoietic tissues and graft versus host disease. The assay further confirms that activation of LFA-1 on NK cells leads to their granule polarization, even if, in some cases, this also takes to an inhibition of NK cell degranulation, suggesting that LFA-1 engagement by ICAMs on target cells may differently affect NK cell response. Finally, we observed that NK cells undergo a time-dependent spontaneous (cytokine-independent) activation after blood withdrawal, an aspect that may strongly bias the evaluation of the resting NK cell response. Altogether our data may pave the way to develop new NK cell activation and expansion strategies that target the highly cytotoxic CD56^dim^ NK cells and can be feasible and useful for cancer and viral infection treatment.

## 1. Introduction

Natural killer (NK) cells are large granular cytotoxic lymphocytes considered part of the innate immune system. They provide rapid responses against virally infected and tumour cells; for these features, NK cells have been successfully employed in several immunotherapeutic strategies [1,2,3,4,5,6]. NK cells express both activating and inhibitory receptors and the balance between their signalling plays an important role in self-tolerance and in NK cell activity [2,7,8]. Healthy cells normally express MHC class I (MHC-I) molecules that, binding to NK inhibitory receptors, grant protection from NK cell activity. On the other hand, “stressed” cells, both exposing ligands for NK cell activating receptors and downmodulating MHC-I molecules, unleash NK cell aggressiveness through a “missing self” recognition mechanism [7,8]. In humans, main actors of this mechanism are HLA class I (HLA-I)-specific inhibitory receptors, killer cell immunoglobulin–like receptors (KIRs) and C-type lectin (CD94-NKG2A), and several NK cell activating receptors [7,8,9]. Interestingly, MHC class I-specific inhibitory receptors allow NK cells to sense not only down modulations in HLA-I expression, but also their own “license” to kill [[10]]. Within the whole NK population, the subset of NK cells expressing NKG2A and/or inhibitory KIRs specific for self MHC class I is statistically more cytotoxic (licensed) than the remaining one (unlicensed).

Human NK cells are classically divided into two subsets based on their level of CD56 expression, namely CD56^bright^ and CD56^dim^ NK cells [[11]]. In peripheral blood (PB), the CD56^bright^ NK cell subset represents 5–10% of the whole population, and upon cytokine stimulation this subset starts to produce high levels of cytokines and becomes highly cytotoxic [11,12,13]. On the other hand, CD56^dim^ NK cells express high levels of cytotoxic proteins and IFN-gamma mRNA and, after target cell recognition, promptly perform spontaneous cytotoxicity and start an abundant IFN-gamma (and other chemokine/cytokine) production [14,15]. CD56^bright^ and CD56^dim^ NK cells have been proposed to represent either NK cells at different stages of differentiation or different NK cell subpopulations [12,13,16,17]. Consistent with this latter hypothesis, there is evidence of their preferential responses to distinct types of stimuli (target- vs. cytokine-mediated activation), of their differentiation from distinct hematopoietic progenitors and of their distinct chemokine driven tissue distribution and trafficking profiles [14,17,18,19,20,21]. As a matter of fact, CD56^dim^ NK cells are preferentially activated after target cell recognition, whereas CD56^bright^ NK cells by NK cell-activating cytokines. Moreover, the two subsets show different effector response to various cytokines as IL-2, IL-15, TGF-beta and IL-21 [14,22,23,24]. Of note, both CD56^bright^CD16^dim/neg^ NK cell subset and unlicensed CD56^dim^CD16^bright^KIR^neg^NKG2A^neg^ NK cells have been described as immature stages of the highly cytotoxic CD56^dim^CD16^bright^ licensed NK cells [10,11,12,25]. Considering that CD56^bright^CD16^dim/neg^ NK cells are virtually all licensed by CD94-NKG2A expression, it is not clear which is the actual sequence of NK maturation steps, suggesting the existence of at least two “converging” pathways of NK cell differentiation and/or that only one of the two subsets (CD56^bright^CD16^dim/neg^ or unlicensed CD56^dim^CD16^bright^) could represent the real immature stage of mature CD56^dim^CD16^bright^ licensed NK cells.

Several receptors have been described to participate in the spontaneous NK cell activation, among them NKp46 (CD335), NKp30 (CD337) and NKp44 (CD336), collectively termed Natural Cytotoxicity Receptors (NCRs), NKG2D (CD314), DNAM-1 (CD226), 2B4 (CD244), LFA-1 (CD11a-CD18) and CD2 are some of the most important [7,8,26,27,28,29]. Activating receptors can recognize either self-ligands, whose expression is usually upregulated upon cellular stress, or exogenous microbial molecules and associate with different intracytoplasmic molecules [2,3,7,30,31].

Among the NCR family, the molecule NKp46 is a major NK cell-activating receptor involved in the fight against bacterial, tumour, and virus-infected cells [30,31,32]. Even if NKp46 was found to recognize some pathogen-associated ligands, a major non-microbial cancer-associated ligand is still lacking and its in vitro activation is still mainly covered by agonistic monoclonal antibodies (mAbs) that mimic the ligand binding. Indeed, anti-NKp46 mediated crosslinking has been described to induce Ca^2+^ flux, triggering of NK cell cytotoxic activity and cytokine release [7,30,31,32]. However, activating receptors can induce natural cytotoxicity only when co-engaged with LFA-1, the major leukocyte adhesion receptor recognizing ICAMs broadly expressed on target cells [33]. Indeed, LFA-1 is essential in leading to granule polarization, which in turn is fundamental for the cytolytic effect [27,34,35,36]. On the other hand, 2B4, CD2 and DNAM-1 molecules work as co-receptors, synergizing with activating molecules [26,27] and targeting NK cell responses against a specific (normal or pathological) cell/tissue expressing their ligands. 2B4 might direct NK cell response against/towards hematopoietic cells by binding CD48 molecule [27,36,37,38], while DNAM-1 against/towards DCs, endothelial cells or specific tumour histotype by binding its ligands, poliovirus receptor (PVR) and Nectin-2 [39,40,41,42]. CD2 (LFA-2), expressed by all CD56^bright^ NK cells and by a fraction of CD56^dim^, is an adhesion molecule which binds CD58 (LFA-3), another adhesion molecule broadly expressed on both hematopoietic and non-hematopoietic cells [43]. CD2 engagement by LFA-3 is involved in NK cell activation [7], suggesting that it may function as co-receptor during natural cytotoxicity.

Even if, as already mentioned, LFA-1 engagement mediates NK cell granule polarization [27,34,35,36], it has been also shown to inhibit NK cell functions inducing both lower percentages of degranulating NK cells [27,35,36] and apoptosis of activated NK cells. For these reasons, LFA-1 engagement has been also proposed as a negative feedback regulatory mechanism to keep NK cell cytotoxic function under control [44].

On account of what was previously said, it is clear that target cell recognition by NK cells in both (patho)physiological and immunotherapeutic conditions is a highly dynamic process controlled by integrated signals coming from multiple receptors that can promote or inhibit adhesion, granule polarization and degranulation. As a matter of fact, a response by resting CD56^dim^ NK cells can be induced by combinations of activating and co-activating receptors, and no single receptor alone has been described to produce significant cytotoxic response [26,27]. However, despite of all the acquired knowledge, it is still unclear whether different NK cell subsets have a similar behaviour or differ in their responses. Aim of the present work is to define, in various NK cell subpopulations (CD56^bright^ vs. CD56^dim^ NK cells and licensed vs. unlicensed CD56^dim^ NK cells), the synergy of NKp46 stimulation with different co-activating signals during NK cell degranulation process with the final goal of paving the way for possible improvements in NK cell mediated immunotherapies.

## 2. Materials and Methods

### 2.1. Monoclonal Antibodies

MAbs for flow cytometry were from different companies (see Table 1 for specifications). Briefly, mAbs to CD16 (FcγRIII, clone 3G8), CD3 (CD3 ε-chain, clone UCHT1), CD45 (LCA, clone C11) were from Ancell Corporation (Bayport, MN, USA), to CD56 (NCAM, clone MEM-188), CD107a (LAMP-1, clone H4A3) were from BioLegend (San Diego, CA, USA), to CD158a/h (KIR2DL1/S1, clone HP-3E4), CD158b/j (KIR2DL2/3/S2, clone CH-L) were from BD Biosciences (San Diego, CA, USA), to CD18 (LFA-1 β2-subunit, clones MEM-148 and MEM-48) were from Immunological Sciences (Rome, Italy), to CD159a (NKG2A, clone Z199) was from Beckman Coulter (Miami, FL, USA), to CD158e (KIR3DL1, clone DX9), CD335 (NKp46, clone 9E2, functional grade) and CD2 (LFA-2, clone LT2, functional grade) were from the human NK Cell Activation/Expansion Kit (Miltenyi Biotec, Bergisch Gladbach, Germany) to CD226 (DNAM-1, clone DX11) was from AbCam (Cambridge, UK) and to CD244 (2B4, clone C1.7) was from eBiosciences (San Diego, CA, USA).

### 2.2. Cell isolation and Culture

Human peripheral blood mononuclear cells (PBMC) were obtained by density gradient centrifugation (Ficoll/Histopack-1077, Sigma-Aldrich, St Louis, MO, USA) of PB samples obtained from healthy donors (Transfusion Centre of Urbino Hospital). To minimize NK cell manipulation/activation, no monocyte adherence was performed. Samples used for the analysis of licensed vs. unlicensed CD56^dim^CD16^bright^ populations were obtained from HLA-I typed donors. Experiments were performed at time 0 (resting NK cells) or at various time points after cell culture with or without human recombinant IL-2 (20ng/mL, specific activity: 1×10^6^ IU/mg, Miltenyi Biotec) in RPMI 1640 supplemented with 2mM L-glutamine, penicillin, streptomycin and 10% fetal bovine serum (BioWittaker, Walkersville, MD, USA) at 37 °C with 5% CO_2_.

### 2.3. Microbead Stimulation of NK Cells

In order to stimulate NK cells, anti-biotin coated microbeads (MACSi Beads™ Particles, Miltenyi Biotec) loaded with biotinylated (agonistic) antibodies directed against various NK cell receptors (total mAb amount 200 ng per 10^6^ microbeads) were used. The microbeads (size: 4–5 µm) mimic the interaction of NK cells with cell targets, leading to NK cell activation. The anti-biotin microbeads were loaded with different combinations of biotinylated agonistic antibodies directed against the following receptors: NKp46 (clone 9E2) [45], 2B4 (C1.7) [26,46], DNAM-1 (DX11) [26,28,29], CD2 (LT2, from human NK Cell Activation/Expansion Kit, Miltenyi Biotec) [47] and LFA-1 β2-subunit (CD18, the signal-transducing subunit of LFA-1 adhesion molecule). For what concern LFA-1 stimulation, MEM-48 and MEM-148 anti-LFA-1 β2-subunit activating clones were used. MEM-48 clone is a functional mAb able to increase cell adhesion, recognizing an epitope expressed on resting leukocytes [48,49]. Differently, the LFA-1 β2-subunit epitope recognized by MEM-148 clone is exposed when LFA-1 is in its high-affinity state and is able to induce the high-affinity conformation of LFA-1 (typical of activated leukocytes) from its low affinity one [50,51]. Unloaded microbeads and/or microbeads loaded with anti-CD56 mAb were also used as negative controls (spontaneous degranulation).

### 2.4. CD107a Degranulation Assay

Cell degranulation was indirectly measured by measuring CD107a surface expression [52,53]. Both percentages of CD107+ degranulating cells (relative number of activated NK cells) and CD107a MFI (mean fluorescence intensity) of degranulating NK cells (proportional to the number of exocyted granules per cell) were evaluated.

Anti-Biotin microbeads (Miltenyi Biotec) loaded with various combinations of biotinylated mAbs and anti-CD107a PerCP-Cy5.5 were added to 10^6^ PBMC in 100 µL final volume of complete medium (RPMI plus 10% FBS) and gently centrifuged (5′ at 500 r.p.m.) to favour NK cells and microbeads contact. Samples were then incubated for 2 h at 37 °C in 5% CO_2_.

After a two hour incubation, cells were stained (20′ at 4 °C) with mAbs to NK cell markers. mAbs to CD16 and to CD56 antigens were used to distinguish between CD56^dim^ and CD56^bright^ NK cell subsets, while anti-KIR (CD158a, b and e), anti-NKG2A and anti-CD16 mAbs to depict licensed and unlicensed NK cells within the CD16^bright^CD56^dim^ NK cell population. Cells were then washed and analysed by flow cytometry, aliquots of some samples were also evaluated by confocal microscopy and Amnis ImageStream (Luminex, Austin, TX, USA).

At first, agonistic mAbs to NKp46, 2B4, CD2 and DNAM-1, anti-LFA-1 β2-subunit were tested individually. Among them, only anti-NKp46 mAbs was able to induce significant and reproducible percentages of CD107a^+^ degranulating NK cells. As expected, combinations of anti-CD2, -DNAM-1 and -2B4 mAbs were unable to induce NK cell degranulation (see Figure 1), but they influenced the degranulation process when associated to anti-NKp46. Using the above described mAb loaded microbeads, we tested different effector/bead ratios. Considered that the stimulation plateau, using microbeads loaded with anti-NKp46 alone, was reached at 1:1 (see Appendix A), this ratio was chosen for all the subsequent experiments when testing the effect of various agonistic mAb combinations. Interestingly, a similar degranulation plateau was described using P815 target cells instead of beads [[36]], suggesting a good reliability of the assay in mimicking NK vs. target cell cytotoxicity. Moreover, NK cell degranulation and granule polarization processes were also tested against Jurkat cell line (positive control). Briefly, Jurkat were stained overnight with 5 µM of the green fluorescent probe DiOC_18_ (D-275, Molecular Probes, Eugene, OR, USA) and then washed [[17]]. As for microbead protocol, Jurkat cells were added to PBMC (1:1 ratio) together with anti-CD107a, centrifuged (5′ at 500 r.p.m.) then incubated for 2 h before performing flow cytometric analysis.

### 2.5. Flow Cytometry

Samples have been acquired on FACScalibur flow cytometer (BD Biosciences) and data analysis was performed using BD CellQuest software program. NK cells were selected within the lymphocyte scatter region (gating out dead cells that possess low forward and relatively high side scatter characteristics, see Appendix A) and then gated on the basis of the CD3^neg^CD56^pos^ phenotype. CD56^dim^ vs. CD56^bright^ or licensed vs. unlicensed NK cells were further characterized by correlate analysis of CD16 expression and CD56 or KIR-NKG2A, respectively. Within the CD16^bright^ population, self-KIR^+^ and/or NKG2A^+^ licensed and KIR^neg^/NKG2A^neg^ unlicensed subsets were defined. Based on CD56 and CD16 intensity of expression, CD56^dim^CD16^bright^(CD3^neg^) and CD56^bright^CD16^dim/neg^(CD3^neg^) were well distinguishable even after 3 days of IL-2 stimulation. Both percentages and mean fluorescent intensity (MFI) of CD107a^+^ NK cells were evaluated. It is well known that CD56^bright^ NK cells possess a lower number of granules than CD56^dim^ NK ones. Indeed, the CD107a MFI of CD56^bright^ degranulating NK cells was significantly lower than that of CD56^dim^ ones [[26,36] and Figure 1], nevertheless the degranulation assay allowed to detect degranulating NK cells in both subsets (see Figure 1). In order to set quadrant markers (degranulation cut-off), PBMC were incubated with mAb unloaded (or loaded with anti-CD56 mAb) microbeads (see Figure 1). The percentage of degranulating NK cells induced by microbeads without antibodies or coated with anti-CD56, usually similar and inferior to 1% (see Figure 1), were subtracted from the experimental values. Percentage of CD107a^+^ NK cells indicated the relative number of degranulating NK cells, while CD107a^+^ MFI was used as surrogate parameter for the amount of exocyted granules per cell. Flow cytometric pictures were generated with WinMDI software and assembled with Adobe Photoshop software (Adobe Systems Inc., San Jose, California, USA).

### 2.6. Confocal Microscopy and ImageStream

In order to evaluate granule polarization during NK cell degranulation, the surface distribution of CD107a antigen was analysed by confocal microscopy and ImageStream. For confocal microscopy observations, a small aliquot (10 µL) of the same samples prepared for flow cytometric analysis (a 2h assay time) was used. Samples, placed on a polylysinated slide, were observed by a Leica TCS-SP5 confocal microscope equipped with DMI 6000 CS inverted microscope (Leica Microsystems CMS GmbH, Mannheim, Germany) and analysed with the Leica Applications Suite Advanced Fluorescence (LAS AF) software. The surface distribution of CD107a was also evaluated on NK cells incubated with Jurkat cells (NK granule polarization positive control). This cell line is a NK sensitive target [[54]] able to activate NK cytotoxic response, thus inducing a contextual polarization of NK granules, necessary for an effective cytolytic activity. To confirm confocal microscopy data, some experiments were performed with an ImageStream device (Luminex, Austin, TX, USA) equipped with a 488 nm laser. ImageStream data were analysed with IDEAS 3.0 software (Amnis Corporation, Seattle, WA, USA).

### 2.7. Statistical Analysis

Percentages and mean fluorescent intensity (MFI) of CD107a surface expression within the different NK cell subsets were calculated as mean values ± their standard deviations (mean ± SD) of at least five independent experiments. Regarding the CD107a MFI of degranulating NK cells, we observed a significant variability from subject to subject, indicating different degranulation capability. For this reason, to evaluate the synergistic effect of CD2, 2B4 and DNAM-1 in NKp46-mediated granule exocytosis and “normalize” our data, the increment of CD107a MFI in each costimulation with agonist mAbs was analysed as percentage increase compared to that induced by NKp46 alone (considered as baseline):(1)$$\frac{\mathrm{mAbs}\;\mathrm{cocktail}\;\mathrm{stimulation}\;\mathrm{value}}{\mathrm{NKp}46\;\mathrm{stimulation}\;\mathrm{value}}\times 100-100$$

In order to evaluate the existence of statistically significant differences, the two-tailed, two-sample Student T test was used. *p* values < 0.05 were considered statistically significant.

## 3. Results

### 3.1. NKp-46 Induced Degranulation in Total Resting PB NK Cells

NK cell degranulation induced by anti-NKp46 mAb in combination with mAbs to CD2, DNAM-1 and 2B4 coactivating receptors was first analysed in peripheral blood resting (day 0) NK cells. Within resting NK cells (the majority of which were CD56^dim^CD16^bright^ licensed NK cells, >80%), percentage of CD107a^+^ (degranulating) NK cells and CD107a MFI of CD107a^+^ NK cells (surrogate marker of the number of exocyted granules) were evaluated. Only stimulation with anti-biotin microbeads loaded with anti-NKp46 alone or in combination with monoclonal antibodies against other activating (2B4, DNAM-1 and CD2) receptors was able to induce significant percentages of degranulating NK cells in all three subsets (see Figure 1 and Appendix A). The lack of NK cell degranulation in the absence of anti-NKp46 stimulus confirmed a co-activating role for CD2, DNAM-1 and 2B4 [55].

All mAbs to coactivating molecules in combination with anti-NKp46 induced an increase of NK cell degranulation, however, among the different stimulatory combinations, only the coactivation with anti-2B4 mAb produced a significant synergistic effect respect to anti-NKp46 alone, independently on other coactivating stimulations (DNAM-1 and CD2). This was evident in terms of both CD107a^+^ NK cell percentage and CD107a MFI on degranulating (CD107a^+^) NK cells (Figure 2A,B and Appendix A). Associated with the intensity of degranulation, we also observed a reduction of CD16 (MFI) expression, indicating that shedding of CD16 from the surface of CD107a^+^ NK cells occurred concomitantly with their degranulation (see Appendix A). We also tested our assay on NK cells purified using the negative immunomagnetic selection system (Miltenyi Biotec) but this technique induced remarkably higher percentages of degranulating NK cells. In a same subject, immunomagnetic NK cell isolation induced an increase of degranulating NK cells from 6% (freshly isolated PBMCs) to 16.5% upon stimulation with anti-NKp46 alone and from 17% (freshly isolated PBMCs) to 36,5% after costimulation with anti-NKp46 plus anti-2B4, respectively. For this reason, we did not use degranulation data obtained from purified NK cells.

### 3.2. Degranulation of Resting PB NK Cell Subsets: CD56^bright^ and CD56^dim^ NK Cells and Licensed and Unlicensed CD56^dim^ NK Cells

The combinations of three mAbs did not produce further synergistic effects when compared to combinations of anti-NKp46 plus one coactivating mAb, for this reason the subsequent experiments were performed only with combinations of anti-NKp46 plus one coactivating mAb.

The NK cell degranulation process was evaluated distinguishing among the following NK cell subpopulation: CD56^bright^CD16^dim/neg^ vs. CD56^dim^CD16^bright^ NK cells and, within CD56^dim^CD16^bright^ NK cells, licensed vs. unlicensed ones. NK cells were first divided in CD56^dim^ and CD56^bright^ subsets on the basis of their CD56 and CD16 MFI (see Figure 1 and Appendix A). As expected, the percentage of CD107a^+^ and the CD107a MFI of CD56^bright^ degranulating NK cells was usually lower than those of CD56^dim^ NK cells and combinations of anti-NKp46 with coactivating mAbs induced an increase of degranulating NK cells in both CD56^dim^CD16^bright^ and CD56^bright^CD16^dim/neg^ subsets (Figure 3 and Appendix A). Of note, a synergistic response of anti-NKp46 mAb in combination with anti-2B4 was confirmed in CD56^dim^CD16^bright^ NK cells (Figure 3 and Appendix A), but not in CD56^bright^ ones (both CD16^neg^ and CD16^dim^, Figure 3 and see Appendix A). Combination of anti-NKp46 with anti-2B4 induced about threefold increase in the number of CD56^dim^ degranulating cells respect to anti-NKp46 alone (NKp46 vs. NKp46+2B4, *p* < 0.05) and the CD107a MFI of CD56^dim^ degranulating NK cells incremented of about 20% respect to NKp46 alone (Figure 3A,B). Differently, coactivation with anti-DNAM-1 or anti-CD2 did not induce statistically significant increases of degranulated NK cell percentage and the increment of CD107a MFI was usually <10% on both CD56^dim^ and CD56^bright^ subsets (Figure 3A,B and Appendix A). CD56^dim^CD16^bright^ cells were further divided into licensed and unlicensed. To determine licensed (expressing NKG2A and/or KIR directed against self HLA-I) and unlicensed NK cells (characterized by a self-KIR^neg^NKG2A^neg^ phenotype), only blood from HLA-typed donors was used. Within the CD56^dim^CD16^bright^ cells, no relevant differences were observable among the different licensed populations (KIR^+^NKG2A^+^, KIR^+^NKG2A^-^, KIR^-^NKG2A^+^) because of subject-dependent variability of response (not shown). As expected, licensed NK cells showed higher degranulation percentages compared to the unlicensed (hyporesponsive) populations (Figure 4A). Similar to total CD56^dim^ NK cells, both subsets showed a significant synergistic response when anti-NKp46 was combined with anti-2B4. Indeed, in both subsets, combination of anti-NKp46 with anti-2B4 induced about threefold increase in the number of CD56^dim^ degranulating cells compared to anti-NKp46 alone (NKp46 vs. NKp46+2B4, *p* < 0.05) and also the CD107a MFI of CD56^dim^ degranulating (CD107a^+^) NK cells remarkably incremented when anti-2B4 was used together with anti-NKp46 (Figure 4A,B and Appendix A).

### 3.3. Degranulation of in Vitro Cultured NK Cell Cubsets: CD56^bright^ and CD56^dim^ NK Cells and Licensed and Unlicensed CD56^dim^ NK Cells

NK cell degranulation induced by stimulation with microbeads loaded with different agonistic mAbs has been evaluated by flow cytometry in cells cultured for 3 days with or without IL-2. As expected, when stimulated with IL-2, NK cells showed significantly higher degranulation percentages than resting NK cells, however, the percentages of degranulating NK cells were not significantly different from NK cells cultured without cytokine stimulation (Figure 5A).

Indeed, along with days of culture, surviving NK cells showed a progressive increase in their degranulation response which was relatively independent on cytokine stimulation (see Appendix A, as an example). As a matter of fact, NK cells cultured for three days with or without IL-2 and stimulated with anti-NKp46 showed degranulation percentages which were more than fourfold higher than those of resting NK cells (Figure 5A and Figure 1). After NK cell activation with cytokines, no statistically significant degranulation differences were detectable in the various NK cell subsets (CD56^bright^ and CD56^dim^ or licensed and unlicensed CD56^dim^ NK cells) for the mAb combinations tested (Figure 5B,C). Moreover, PBMCs stimulated with anti-NKp46 plus anti-2B4 produced a diffuse CD107a labelling pattern on degranulating NK cells (Figure 5D), indicating that this stimulus combination did not induce significant granule polarization.

### 3.4. Granule Polarization of in Vitro Vultured NK Cells and LFA-1-Mediated Inhibition of Degranulation

As expected, stimulation with anti-NKp46 plus anti-coreceptors did not take to significant granule polarization (see Figure 5D). For this reason, we tested the effect of anti-LFA-1 β2-subunit (CD18) agonistic mAbs added to anti-NKp46 loaded microbeads considering that LFA-1 (CD11a-CD18 heterodimer) has been described to be involved in this process [27,34,35,36]. To induce granule polarization, two different anti-CD18 agonistic clones directed against LFA-1 β2-chain, namely MEM-48 and MEM-148, were tested. In order to increase the number of degranulating NK cells within the PBMCs and to allow an easier evaluation of the (relatively rare) polarization process, experiments were performed on cultured PBMCs. Surprisingly, NK cells reacted differently to the two anti- LFA-1 β2-subunit agonistic antibodies, clone MEM-48 significantly inhibited NKp46-induced release of granules either in absence (Figure 6A) or in presence of anti-2B4 (54.0 ± 14.4 vs. 31.7 ± 20.4 with MEM-48, n = 3, *p* < 0,05), while MEM-148 did not affect the degree of degranulation neither in absence (Figure 6A) nor in presence of anti-2B4 (63.0 ± 7.5 vs. 59.3 ± 10.6 with MEM-148, n = 3, ns). Inhibition of NKp46-induced CD107a expression by MEM-48 clone on CD56^bright^ NK cell subset was also evident (38.3 ± 4.2 vs. 28,0 ± 5,0 with MEM-48, n = 3, *p* < 0.05, see also Appendix A). In line with these data, engagement of LFA-1 on resting NK cells by ICAM-1 expressed on transfected SC2 insect cells showed to reduce the percentage of degranulating NK cells induced by activating receptor triggering [27,35,36].

To verify whether the microbeads-based assay was able to properly reproduce a polarized process of degranulation, we analyzed it by confocal microscopy. The pattern of expression induced by microbeads loaded with anti-NKp46 plus anti-2B4 and anti- LFA-1 β2-subunit (clone MEM-148) (Figure 6B) was similar to what observed by co-incubating NK cells with a NK-sensitive cell target, the green-labelled (DiOC_18_) Jurkat cell line (positive control, Figure 6, insert H). Indeed, this cell line is able to activate a NK cytotoxic response, thus inducing both NK granule polarization and degranulation, necessary for an effective cytolytic activity. The juxtaposition of imaging techniques to conventional flow cytometry allowed us to follow at the same time granule localization and degranulation, giving us proof of the polarized degranulation process induced by specific activating and co-activating molecule combinations. Notably, we were able not only to detect polarization of CD107a expression on NK cells, but also trogocytotic transfer of CD107a (red spots) on some Jurkat (green) cells (Figure 6, insert H). Moreover, we observed that the 2h incubation with labelled Jurkat cells induce a transfer of Jurkat green fluorescence to some green negative effector cells, as suggested by both some isolated green/orange/yellow spots (see Figure 6, insert H) and a higher green fluorescence on degranulating CD107a positive than CD107a negative NK cells (see Appendix A). To confirm the confocal microscopy (pure imaging) observations, the same samples were cross-checked with the ImageStream (imaging in flow) technology (see Figure 6C).

## 4. Discussion

The possibility of expanding and harnessing NK cells for cancer treatment led both to the development of NK adoptive transfer immunotherapies and to the use of monoclonal antibodies targeting the NK cell activating and inhibitory receptors [1,2,4,5,6,56,57]. Nowadays several efforts are addressed to study and manipulate NK immune checkpoints, while data on synergistic cooperation induced by different NK surface activating receptors on distinct NK cell subsets are still incomplete.

### 4.1. Assay Sensitivity

Aim of the present work has been to evaluate the synergistic effect of NKp46 with some coactivating receptors on different NK cell subsets (CD56^bright^ and CD56^dim^ NK cells and licensed and unlicensed CD56^dim^ NK cells) in inducing both degranulation and granule polarization processes on human PB NK cell subsets. The ability of an anti-NKp46 mAb to induce degranulation on resting NK cells has been already described [58]; however, its synergistic cooperation with co-activating receptors on distinct NK cell subsets has never been depicted yet. To investigate the NK cell subset responses to activating stimuli we developed a microbead-based assay using different agonistic mAb combinations against NK-activating and -coactivating receptors. The microbeads were chosen in order to mimic the physiological conditions of NK-target cell interactions, as well as to possibly increase assay sensitivity. Indeed, although CD56^bright^ NK cells express lower amounts of intracellular cytotoxic proteins and of CD107a antigen [36,59] and unlicensed NK cells were considered hyporensponsive [10], we were able to induced detectable levels of CD107a expression on both these less-cytotoxic NK cell subsets, indicating a good sensitivity of the assay.

### 4.2. B4 Activating Co-Operation on CD56^dim^ NK Cells

Interestingly, anti-NKp46 preferentially synergized with anti-2B4 on resting CD56^dim^ (both licensed and unlicensed) NK cells, while no significant degranulation differences were observed among the various co-activating receptors on CD56^bright^ cells. The synergistic effect was evaluated in term both of increased number of cytotoxic NK cells and of granule released (cytotoxic capability) per single NK cell. In line with our observation, experiments performed with ligand (for NK cell activating receptors) expression on transfected SC2 insect cells showed that co-engagement of 2B4 and CD16 or NKG2D resulted in enhanced frequency of degranulating CD56^dim^ NK cells [27,36]. Hence, exclusively in this highly cytotoxic NK cell subset, triggering of CD16, NKG2D or NKp46 activating receptors on CD56^dim^ NK cells was significantly increased by 2B4 co-engagement, suggesting a common synergist activation mechanism of 2B4 with all three major NK cell activating receptors. Of note, CD16 and NKp46 are immunoreceptor tyrosine-based activation motif (ITAM)-associated receptors, while NKG2D associates with the adaptor protein DAP10 and 2B4 signal is mediated by immunoreceptor tyrosine-based switch motifs, ITSMs [55,60]. This suggests that, on CD56^dim^ NK cell subset, 2B4-ITSMs can mediate a synergistic cooperation that involves different activating signal transduction pathways and might operate downstream the associated activation motifs/adaptors. Differential expression of 2B4 on surface and/or intracellular adaptors, signalling lymphocyte activation molecule (SLAM)-associated protein (SAP) and Ewing’s sarcoma-associated transcript 2 (EAT-2), between CD56^dim^ and CD56^bright^ NK cell subsets may account for their different behaviour [26,61]. As a matter of fact, surface expression of 2B4 has been depicted at slightly higher density of expression on CD56^dim^ than CD56^bright^ NK cells [26,62]. On the other hand, CD56^bright^ NK cells displayed significantly higher expression of NKp46 (surface density) and CD2 (surface density and percentage of positive cells) respect to CD56^dim^ NK cells [26,62], while there were no significant differences in DNAM-1 expression between the two subsets [26,28,63].

On CD56^dim^ NK cells, the synergistic activity of 2B4 in combination with activating receptors seems to be confined to NK cell degranulation/cytotoxic activity since it was not evident for NK cell cytokine secretion [14].

### 4.3. B4 Preferentially Drives NK Cell Function Against Hematopoietic Cells (Exemplified in Haploidentical Transplantation)

2B4 receptor is triggered by engagement with CD48 antigen, a GPI-anchored surface molecule constitutively (and exclusively) expressed on nearly all lympho-hematopoietic cells and endothelial cells [64]. We have already suggested that NK co-receptors may drive tissue specific NK-mediated cytolysis. To this regard, the restricted expression of CD48 to hematopoietic cells, sets limits to 2B4-dependent synergistic cytotoxic activity of CD56^dim^ NK cells suggesting that CD56^dim^ NK cells may represent a cytotoxic effector subset deeply involved in patrolling hematopoietic compartment. Indeed, allogeneic hematopoietic stem cell transplantation (HSCT) from selected donors showed to generate alloreactive CD56^dim^ NK cells able to kill not only patient’s neoplastic cells, but also his dendritic cells (DCs) and T lymphocytes, reducing leukemic relapse, thus graft versus host disease (GvHD), and graft rejection [1,2,3,4]. The observation that CD48-hematopoietic antigen can synergistically triggers CD56^dim^ NK cell cytotoxicity possibly provides a further rational for the lack of NK mediated graft versus host disease (GvHD) in KIR ligand-mismatched haploidentical hematopoietic transplantation. Notably, it has been shown that CD48 expression increases under inflammatory conditions such as during viral and bacterial infections, regulating target cell lysis and viral clearance mediated by cytotoxic cell lymphocytes [64]. Notably, during allogeneic HSCT, patients are exposed to a marked immunosuppression, which is an important cause of viral reactivation/infection. Since viral infection/reactivation in HCST recipients is associated with high morbidity and mortality, adoptive T and NK cell therapy strategies and/or administration of immune activating cytokines have been pursued [56,57,65,66,67]. Indeed, reactivations of Epstein-Barr virus (EBV) as well as cytomegalovirus (CMV) and other herpes viruses can occur more frequently after HSCT [65,66,67] and NK cells are one of the main immune cells fighting these viruses. After primary infection, the EBV virus persists lurking dormant inside B cells [68] and EBV infected B cells are particularly sensitive to CD244-mediated NK cell lysis [[64]]. On the other hand, CMV latent infection is carried out within myeloid and dendritic cell progenitors and virus reactivation can be triggered by growth factors associated with the inflammatory response [69], a condition known to induce CD48 upregulation, thus providing the rational for antigen-independent CD56^dim^ NK-mediated EBV/CMV virus control. Therefore, in haploidentical KIR-ligand mismatch settings, 2B4-mediated CD56^dim^ NK killing of CD48 positive hematopoietic cells could be not only involved in driving CD56^dim^ against host hematopoietic cells and thus (sparing other tissues) limiting GVHD, but particularly against host (and donor) hematopoietic cells undergoing CD48 upregulation due to viral reactivation. In line with this hypothesis, it is known that patients with NK-type chronic lymphoproliferative disease of granular lymphocytes (NK-LGLs) can develop autoimmune disorders, such as cytopenias, vasculitic syndromes and neuropathy [70], suggesting that autoaggressive NK-LGLs might specifically target hematopoietic cells and endothelial cells known to express CD48 antigen. Remarkably, in the large majority of NK-LGL patients, there is a serologic evidence of past viral infection and some patients suffer from recurrent infections related to neutropenia, suggesting that viral infection and subsequent proinflammatory cytokine secretion may play a role in disease pathogenesis [71]. Furthermore, it is tempting to speculate that viral-mediated CD48 upregulation might be in some patients one of the triggers for both NK-LGL expansion and autoimmune disorders.

### 4.4. B4 Signalling on Unlicensed CD56^dim^ NK Cells

As expected, within the CD56^dim^ NK cells, the percentages of degranulated licensed NK cells were significantly higher than those of unlicensed ones in all stimuli combinations, while no relevant difference of degranulation was observed between KIR- and NKG2A-lincensed NK cells. However, depending on the subject, a considerable variability of response was observable between KIR- and NKG2A-lincensed NK cells. A similar subject-dependent variability of response was detectable after co-activation, suggesting that NK cell responses importantly differ from subject to subject. Nevertheless, we were able to observe a consistent synergistic co-operation of 2B4 and NKp46 receptors on both licensed and unlicensed CD56^dim^ NK cell subsets, indicating a common mechanism of CD56^dim^ activation which was not evident for the CD56^bright^ NK cells. Besides the presence of self-KIR and/or NKG2A and the higher CD94 intensity of expression (because of its coexpression with NKG2A on licensed NK cells), DNAM-1 density of expression (in terms of MFI, proportional to the amount of the marker per cell) was the only marker with a significant lower expression on unlicensed respect to licensed NK cells, while both perforin and granzyme B have similar MFI on both subsets [10]. Nevertheless, we did not detect significant reduction of DNAM-1 synergistic activity between unlicensed and licensed NK cells, as compared to CD2 coactivating stimulus (see Figure 4A,B).

### 4.5. CD56^dim^ and CD56^bright^ Converging Phenotypes and Functions Possibly from Different Precursors

We have already highlighted that there is no univocal consensus regarding the actual sequence of NK maturation steps and we and other groups have already suggested the existence of at least two independent “converging” pathways of NK cell differentiation: thymus/lymph node (CD56^bright^) and bone marrow/spleen (CD56^dim^) [17,72]. Indeed, there are several bodies of evidence suggesting that CD56^dim^ and CD56^bright^ cells could represent two distinct NK cell subsets with “converging” phenotypes and function. In line with this, our data show that CD56^dim^ both licensed and unlicensed, but not of CD56^bright^ NK cells, synergistically respond to 2B4 and NKp46 co-triggering, suggesting that unlicensed CD56^dim^ (and not CD56^bright^) NK cells might represent an immature stage of more mature/cytotoxic licensed CD56^dim^ NK cells. Indeed, a proportion of unlicensed NK cells has been shown to start to acquire licensed phenotype expressing KIR-NKG2A receptors after NK stimulating culture conditions [25,73], suggesting that they represent an immature stage of highly cytotoxic CD56^dim^ licensed NK cells. On the other hand, CD56^bright^ cells has been described as specialized NK cells that preferentially respond by mean of their cytokine supply rather than by their cytotoxic activity, contradicting that they represent an immature NK cell stage [58]. Moreover, CD56^bright^ and CD56^dim^ NK cell subsets show substantially different activation pathways (CD56^bright^ by cytokines, CD56^dim^ by targets), respond differently to cytokine stimulation and exhibit distinct chemokine-driven tissue distribution and trafficking profiles [14,20,22,23,24]. To this regard, it has been recently characterized a novel CD34^+^DNAM-1^bright^CXCR4^+^ NK precursor that gives rise to mature “licensed” NK cells, further suggesting the possibility of concomitant NK cell development from different CD34^+^ progenitors [21]. Indeed, using different chemokine/cytokine receptor expression and different end-organ trafficking [20,21,74,75,76,77], we can draw an association between CD56^dim^ NK cells and CD34^+^DNAM-1^bright^CXCR4^+^ (CD117^neg^) NK precursors (both preferentially expressing CX3CR1 and CXCR1), while CD56^bright^ NK cells can be associated to CD34^+^DNAM-1^neg^ NK progenitors (both preferentially expressing CD62L and CD117^+^). CD117 antigen (c-kit receptor) had been already indicated as an exclusive marker of CD56^bright^ NK cell subset [78] and proved to be one of the best way to detect CD56^bright^ NK cells and to distinguish them from CD56^dim^ NK cells when they are developed in vitro from PB CD34^+^ hematopoietic progenitors [17]. Indeed, in vitro developing CD56^bright^ and CD56^dim^ NK cells from CD34^+^ hematopoietic progenitors show a distinct pattern of antigen expression and different mechanisms to prevent NK cell-mediated auto-aggression during their differentiation, suggesting a distinct origin of CD56^bright^ and CD56^dim^ NK cell subsets and two convergent pathways of NK cell differentiation ([17] and L. Zamai unpublished data).

### 4.6. NK Spontaneous Activation After Blood withdrawal and Cytokine Effect

During NK cell culture (1-3 days) in the presence of IL-2, the percentages of NK cells able to degranulate upon microbead stimulation progressively increased and the differences between subsets (e.g., licensed vs. unlicensed) or among different co-stimulations tended to be lost. Engagement of CD122/CD132 cytokine receptor remarkably enhanced degranulation of all NK cell subsets, predisposing a higher proportion of NK cells to degranulation and/or promoting the survival of “prime/preactivated” NK cells more prone to degranulation. Similar to IL-2-activated NK cells, resting licensed NK cells are known to be more prone to degranulate than resting unlicensed ones, suggesting that the presence of self-HLA-I (maybe through an indirect mediated engagement with HLA-I-inhibitory receptors in *cis*) on licensed CD56^dim^ NK cells may predispose a “prime/preactivated” NK cell state similar to that mediated by Ly49 in murine model [79]. Indeed, the hyporesponsiveness of unlicensed state can be “reversed” when NK cell activating receptors are engaged by activating ligands (e.g., CD122/CD132 after IL-2/IL-15 stimulation), finally reducing the degranulation differences between licensed and unlicensed NK cells. Of note, the percentages of degranulating NK cells increase during time of culture even in the absence of IL-2, growth factors present in the culture medium (namely fetal bovine serum, FBS) may predispose NK cell to degranulation and/or the preferential survival of “preactivated/primed” NK cells might “select” NK cells more prone to degranulate. However, even a quick NK purification step through a negative selection induced higher percentages of degranulating NK cells (not shown), suggesting that NK cells spontaneously and rapidly decrease their threshold of activation after their *ex-vivo* manipulation and withdrawal from circulation.

Since only a minor proportion of resting NK cells degranulate, there is a large proportion of licensed NK cells that do not degranulate, while, although few, some unlicensed NK cells are able to degranulate, suggesting that the final outcome (degranulation or not) is determined by a signal integration depending on both “internal predisposition” (e.g., licensing condition) and exogenous microenvironment condition/stimulation (e.g., engagement of cytokine receptors); conditions that may be totally incidental and driven by sequential presence or absence of occasional (activating or inhibitory) receptor/ligand bindings. This the reason why we decided to reduce manipulation steps and avoided doing not only the purification step but also monocyte adherence (see MM).

### 4.7. LFA-1 Mediated Inhibition of NK Cell Degranulation: Multi-Functional Receptors

As expected, the combination of anti-NKp46, anti-2B4 and anti- LFA-1 β2-subunit (CD18, clone MEM-148) mAbs loaded on microbeads was able to reproduce a pattern of NK granule polarization and degranulation similar to that observed with Jurkat cells. Intriguingly, we and others [27,35,36] have observed that, in some cases (e.g., using MEM-48 clone), the engagement of LFA-1 inhibited NK cell degranulation on either cytokine activated and resting NK cells, respectively. We also observed that this occurs on both CD56^dim^(LFA-1^bright^) and CD56^bright^(LFA-1^dim^) NK cells, regardless their differential expression of LFA-1 surface density [27,77]. Indeed, differences among ICAM members in inducing NK cytotoxic activity have been already described [80], suggesting that LFA-1 signals may vary from activating to inhibitory depending on ICAMs engaged and/or NK cell “predisposition”. To this regard, it has been shown that leukotoxin, a protein toxin secreted from an oral bacterium, can target LFA-1 β2-subunit on activated white blood cells, causing their cell death [81]. Moreover, LFA-1 β2-subunit-mediated endothelial(HUVEC)- and tumour (K562) cell-induced apoptosis of activated NK cells has been also described [44,82]. Target cell-induced NK apoptosis mediated by LFA-1 β2-subunit has been suggested as a feedback regulatory mechanism to keep activated NK cell expansion under control [44,82]. A death pathway that could be potentially exploited by cancer cells. Indeed, some studies report that the increased expression of ICAM-1 on tumour cells correlates with a more aggressive cancer [83].

Of note, multi-functionality is not unique to the LFA-1-ICAM-1 interaction. There are other membrane-expressed receptor-ligand pair molecules with opposing function (e.g., KIR/NKG2-HLA-I pairs) and even 2B4-CD48 pair can mediate dual function depending on their association with different adaptor molecules (e.g., presence or absence of intracytoplasmic SAP) [46,84,85,86,87], thus revealing how the immune system has evolved to use even a single molecule in multiple ways depending not only on its integrated interaction with intracellular adaptors, but also with surface molecules. For example, PD-1 inhibitory function is mediated by its association with CD28 and it has been shown that antagonist anti-PD-1 (non-blocking the PD-1/PDL-1 interaction) mAbs are able to block CD28-PD-1-association finally unleashing NK cell activity in the presence of PD-1/PDL-1 engagement [88].

### 4.8. Cis Receptor/Ligand Interactions as Regulatory Mechanism for Receptor Multifunctionality

Dual function of a single receptor-ligand pair has been described to be regulated by receptor cell surface density and affinity, degree of triggering (amount of ligand expression), and the relative abundance of some adaptor molecules, finally mediating signals that can take to opposite response [86]. Interestingly, some surface receptors have been shown to interact with their ligand on the surface of the same cell (the so called *cis* association). This *cis*-association sequestrates receptors thus reducing their surface expression and modulating the receptor function engaged in *trans* by neighbouring cells. For example, sequestration of Ly49A through *cis* association with MHC-class I was shown to license mouse NK cells during their maturation by reducing the inhibitory effect of unengaged Ly49 receptors and increasing MHC-I down-modulation sensing on target cells [79]. *cis-* Interactions have been demonstrated not only for some inhibitory NK cell receptors with their MHC ligands [89], but also for the 2B4 NK cell activating receptor with its ligand CD48 [87]. Indeed, 2B4-CD48 *cis*-interaction has been shown to modulate 2B4 cell surface expression, baseline phosphorylation and cytotoxicity induced after contact with susceptible target cells, suggesting a general role for *cis* interactions in modulating the threshold for receptor engagement in *trans* and the regulation of receptor function [87]. *Cis* interactions can increase or decrease the threshold of NK cell activation depending on receptor type. As already mentioned, it facilitates NK cell activation, as the number of inhibitory receptors available to interact in *trans* is reduced (e.g., mouse Ly49 receptors); by contrast, 2B4-CD48 *cis* interaction dampens NK-cell activation. Interestingly, the acquisition of activation receptor ligand by trogocytosis has been described to dampen NK cell response [90], suggesting that *cis* association of activating receptor (Ly49H) with its ligand (m157) might drive the lack of NK cell activating signals (NK inhibition).

Structural flexibility of extracellular domains has been claimed to be necessary to allow receptor/ligand cell surface interaction in both *cis* and *trans* [79,87,89]. However, hypothetically, sequestration of receptor might potentially occur also in the intracytoplasmic compartment where *cis* interaction might require different structural arrangement and intracellular adaptors. If this is the case, it is possible that some *cis* interactions might not be unveiled yet and *cis*–*trans* interactions of receptor/ligand might occur more broadly than we believe on the base of the scientific evidence. For example, ICAMs, which are constitutively expressed at low levels and readily increase their density of expression under inflammatory cytokine stimulation on a wide range of cells, might give rise to *cis* interaction with LFA-1 on leukocytes. Because ICAM function is regulated primarily by its surface expression level, *cis* interaction could both modulate ICAM surface expression and dual (activating or inhibiting) functions of LFA-1 receptor. Finally, *cis* interactions mediated by intracellular or surface “adaptors” might explain human NK cell licensing mechanism mediated by self-KIR and NKG2A molecules, as functionally analogous mouse Ly49 molecules.

## 5. Conclusions

The possibility to expand and activate NK cells is a crucial aspect to benefit of NK cell functions in immunotherapeutic approaches. Indeed, ex-vivo activated NK cells can be exploited to counter cancer cells after their injection; however, we have only limited knowledge regarding the activating and inhibitory signals induced in the different NK cell subsets. In the present work, we have identified synergistic and antagonistic activation activities on specific NK cell subsets of 2B4 and LFA-1 receptors, respectively. Of interest for cancer therapy, our data indicate that 2B4 triggering was able to act synergistically with NKp46 receptor on CD56^dim^ NK cell subset. Therefore, synergistic costimulation involving 2B4 coreceptor and NK cell activating receptors could be exploited *ex-vivo* to activate/expand the highly cytotoxic CD56^dim^ NK cell subset either through agonistic mAbs to 2B4 receptor or using target/feeder cells expressing CD48 ligand at high level. On the other hand, exploitation of CD48 ligand upregulation induced by cytokine administration on target cells could be hypothesized to induce a higher CD56^dim^ NK cell response/expansion in vivo. To this regard, it is well known the ability of B lymphoblastoid cell lines to act in vitro as feeders that sustain the preferential proliferation of NK cells from PBL mediated by both IL-2 and cell-to-cell contact [[91,92]]. As a matter of fact, an exponential growth of (CD16+) NK cells can be typically obtained by co-culturing PBL with irradiated RPMI-8866 or Daudi or 721.221 lymphoblastoid cells [[91,92]], which, notably, are all three positive for EBV viral genome [93]; a virus that has been shown to induce not only CD48 upregulation [64,94,95] but also the expression of a CD48 activation epitope at the surface (possibly unmasked through association/dissociation from other cell surface molecules) of EBV-transformed B lymphoid cells [94]. Epstein-Barr latent membrane protein 1 (LMP1), among other viral proteins, is a virus-encoded oncogene required for B cell transformation that plays a central role in inducing NF-kB activity and transactivation of the CD48 NF-kB binding site [95], finally providing the rational for CD48 protein upregulation in human EBV-infected B lymphoid cells. Of interest, during acute EBV infection, a selective and persisting (months) expansion of CD56^dim^/NKG2A^+^ or CD16^+^/NKG2A^+^ NK cells has been shown to occur in peripheral blood [96,97,98,99]. This NK cell subset preferentially degranulates and proliferates after exposure to lytic EBV-infected B cells [96], suggesting a specific involvement of a subset of CD56^dim^ NK cells, possibly with memory-like properties, in EBV-mediated NK cell expansion. Similarly, in the presence of influenza virus, the percentage of degranulating (CD107a^+^) NK cells significantly increased and the CD56^dim^ NK cell subset accounted for most of the degranulation [62]. Interestingly, expression of 2B4 on NK cells was upregulated in response to the virus and particularly on CD56^dim^ rather than CD56^bright^ NK cells. Moreover, in response to the virus, 2B4^bright^CD56^dim^ NK cells had a higher degranulation activity than 2B4^dim^CD56^dim^ NK cells, suggesting that 2B4 upregulation could increase NK cell response to influenza [62]. Indeed, the major PB population that can be infected by influenza virus, monocytes, upregulated CD48 surface expression, when challenged by influenza virus, suggesting a direct correlation between 2B4 upregulation and increased NK functional activity [62]. Both 2B4 upregulation and its synergy with NKp46 (a key activating receptor which can interact with influenza haemagglutinins [100]) has been shown in response to influenza virus [62]. Finally, influenza virus induced expansion of both 2B4^+^ NK cells (and this correlates with CD48-upregulation on monocytes) [62] and long-lived NK cells homing in the bone marrow [3]. Long-lived NK cells induced by influenza have been shown to respond also to respiratory syncytial virus challenges. Moreover, after annual flu vaccination, NK cells showed to provide a vaccine-induced wider virus coverage for several months [3], suggesting an antigen independent long-lived NK cell generation that resembles the one induced after cytokine activation [3]. These evidences together with our findings indicate that a strong engagement of 2B4 in trans on NK cell surface (by increased CD48 and/or its activation epitope expression on NK target cells) could represent a crucial mechanism to induce a specific proliferation and expansion of a subset of highly cytotoxic CD56^dim^ NK cells that could acquire an antigen-independent memory-like features similar to those of long-lived cytokine-activated NK cells [3]. To this regard, 2B4 expression has also been associated with a subset of memory CD8^+^ αβ T cells [101,102,103].

In NK-cell based immunotherapy maximal NK cell activation is usually pursued; however, the higher is the activation and aggressiveness of NK cells, the higher will be the upregulation of mechanisms aimed to keep them under control either through immune checkpoint inhibitory receptors or through activation induced cell death mediated by death receptors such as TNF receptor family members or LFA-1 [6,17,44,82], which finally damp NK cell effectiveness. Moreover, although the maximal activation of the highly cytotoxic CD56^dim^ NK cells could be exploitable against tumour cells, it should be kept in mind that an excessive NK stimulation can easily lead to detrimental autoaggressive reactions [104]. Thus, to improve safety, it will be very important to reach an optimal balance between positive and negative actions of activated NK cells. To this regard, a deeper understanding of NK cell biology will not only be crucial for immunotherapeutic NK-based treatments (anticancer and anti-infectious therapies) but it will also enable a safe use of these therapies. Of note, contrasting outcome from different subjects using the same therapeutic protocols have been described, reflecting individual variations in immune (NK) system and tissue characteristics, therefore it will be crucial to shed light on all the main factors involved in NK cell final response which is related to a subject-specific “environmental” context. The increase of our understanding regarding the NK patho/physiologic mechanisms and their interplay with surrounding microenvironment will allow to develop methods to produce activated NK cells able to “surgically” direct their deadly kiss against the designated targets for a more and more personalized medicine. To this regard, we and others have shown a trogocytotic activity between NK cells and their targets [see Figure 6, insert H, [90,105,106]. Indeed, the transfer of (cytotoxic) components from effectors to target during cytotoxic activity is not a unidirectional mechanism, increasing evidence suggests that target cells are equally able to deliver some of their components to effector cells. As a matter of fact, trogocytosis seems a common aspect of activated NK cells and the uptake of membrane lipids and associated molecules from target to effector cells has been shown to modulate NK cell functions by inducing either inhibition or activation or new migratory properties depending on the acquired target molecules [90,105,106,107]. Therefore, this mechanism could be exploited, creating new specific culture conditions in order to both expand NK cells and modulate their aggressiveness. Among the molecules to “trogocytically” deliver to NK cells, ligands for NK receptors or receptors for NK ligands because of their modulatory function in receptor/ligand *cis* interactions and/or cancer-homing receptors will certainly be optimal candidates for these purposes. The improvement of our understanding of NK cell biology can pave the way for designing new NK-based immunotherapies able to harness the most suitable NK cells to counter tumour cells and other diseases.

## Figures and Tables

**Figure 1 cells-09-00753-f001:**
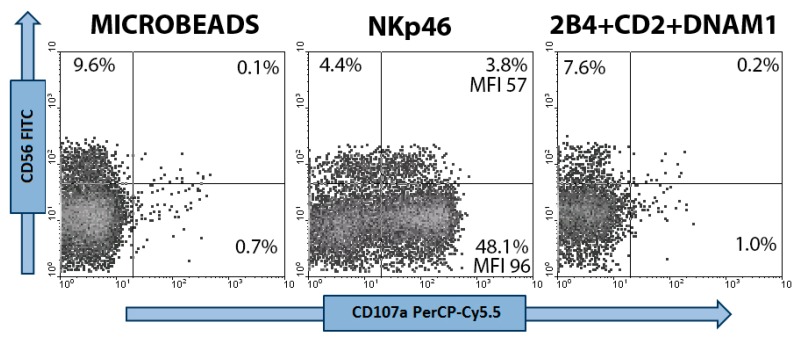
Flow cytometric dot plot analyses of peripheral blood NK cells. Three day IL-2 (1 ng/mL) stimulated NK cells were selected within the lymphocyte scatter region and then gated on the basis of CD56+/CD3- phenotype. NK cell samples were stimulated for 2 h with mAb unloaded microbeads (negative control), or microbeads loaded with anti-NKp46 or with anti-2B4 plus –CD2 and –DNAM-1 mAbs. CD56^dim^ and CD56^bright^ NK populations are distinguished based on CD56 intensity of expression. NK cell degranulation is detected by measuring surface expression of CD107a. Relative percentages of degranulating (CD107a+) NK cells and CD107a MFI are shown.

**Figure 2 cells-09-00753-f002:**
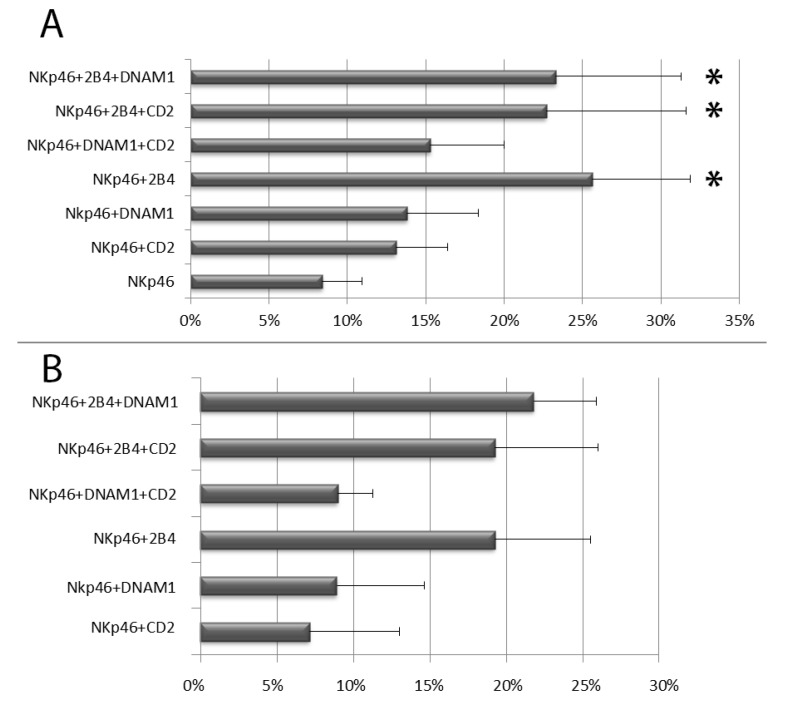
Degranulation analysis of resting peripheral blood NK cells after stimulation with the indicated mAb combinations. (**A**) Percentages of degranulating NK cells. (**B**) Percentage increment of CD107a mean fluorescent intensity on degranulating NK cells stimulated with the indicated combinations respect to anti-NKp46 alone (used as reference, see MM section where the method for calculation of increments is discussed). Data represent mean +/− SD of at least 5 experiments. Bars indicate SD. * *p* < 0.05 relative to NK cells stimulated with anti-NKp46 mAb-coated beads.

**Figure 3 cells-09-00753-f003:**
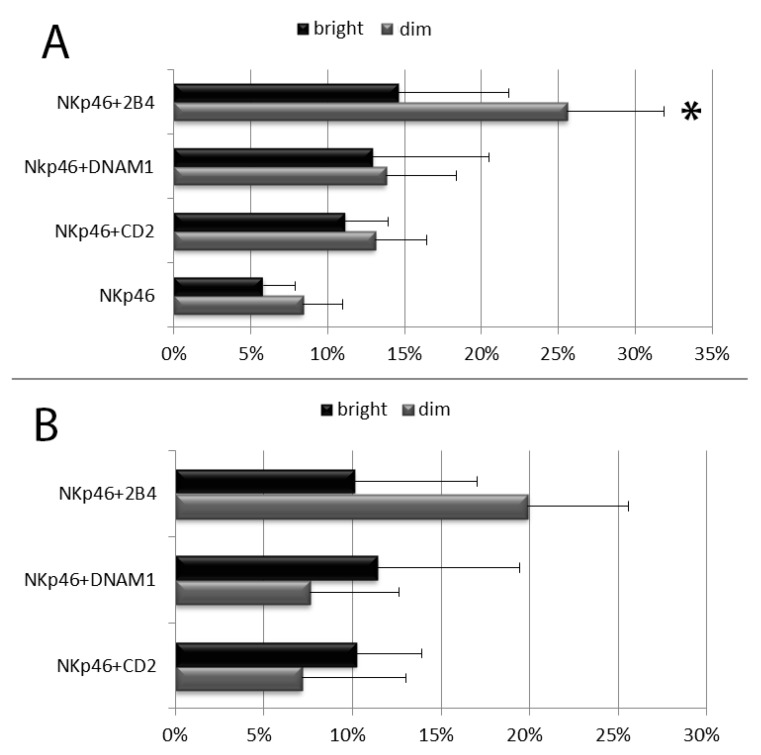
Degranulation analysis of resting CD56^dim^ and CD56^bright^ NK cells after stimulation with the indicated mAb combinations. (**A**) Percentages of degranulating NK cells. (**B**) Percentage increment of CD107a mean fluorescent intensity on degranulating NK cells stimulated with the indicated combinations respect to anti-NKp46 alone. Data represent mean +/− SD of at least 5 experiments. Bars indicate SD. * *p* < 0.05 relative to NK cells stimulated with anti-NKp46 mAb-coated beads within its NK cell subset.

**Figure 4 cells-09-00753-f004:**
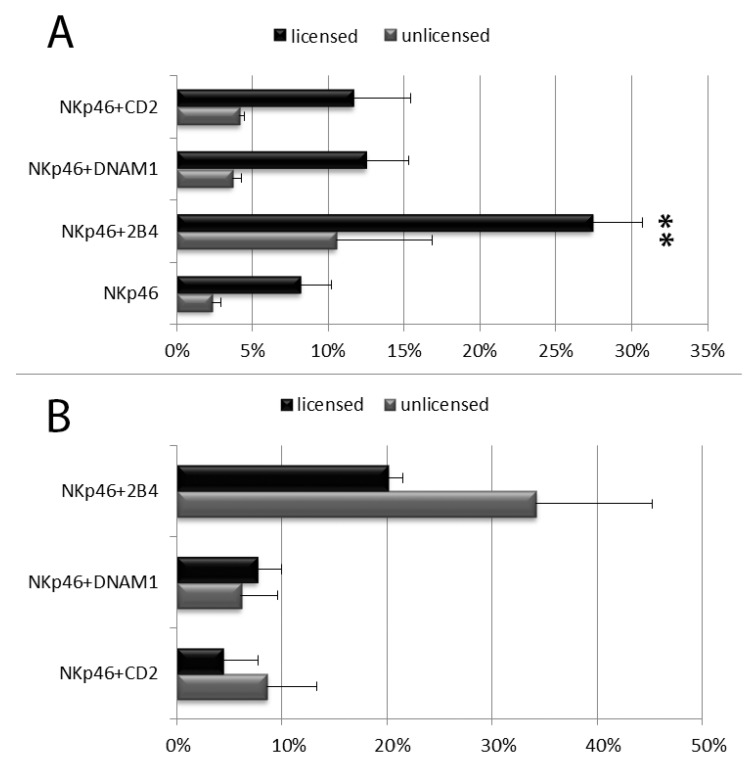
Degranulation analysis of resting licensed and unlicensed CD56^dim^ NK cells after stimulation with the indicated mAb combinations. (**A**) Percentages of degranulating NK cells. (**B**) Percentage increment of CD107a mean fluorescent intensity on degranulating NK cells stimulated with the indicated combinations respect to anti-NKp46 alone. Data represent mean +/− SD of at least 5 experiments. Bars indicate SD. **p* < 0.05 relative to NK cells stimulated with anti-NKp46 mAb-coated beads within its NK cell subset. (** They are two different *, referred to two different close columns).

**Figure 5 cells-09-00753-f005:**
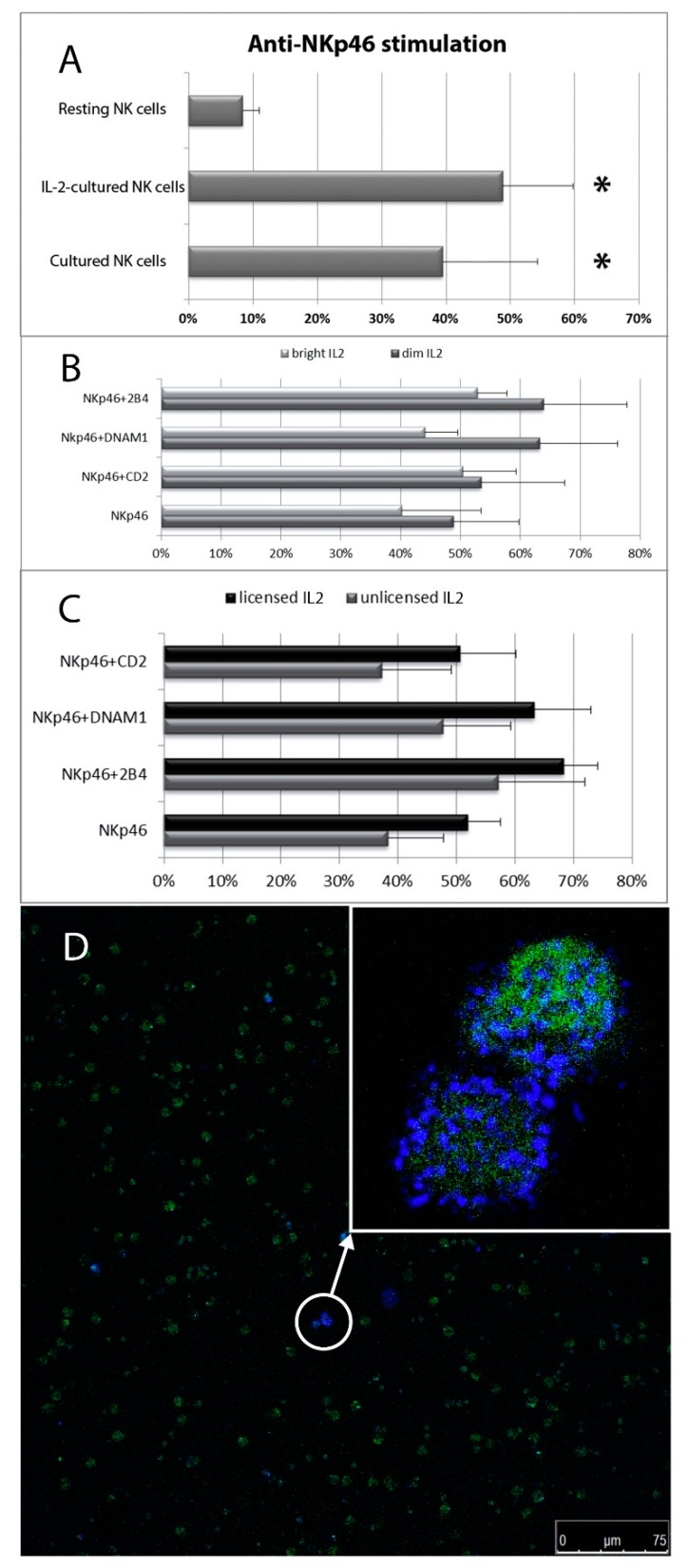
Degranulation analysis of in vitro cultured NK cells. (**A**) NK cell degranulation percentages induced by anti-NKp46 stimulation under three different culture conditions. (**B**) Degranulation percentages of IL-2-cultured NK cells stimulated with various mAb combinations was evaluated by comparing CD56^dim^ and CD56^bright^ degranulating NK cells or (**C**) licensed vs. unlicensed CD56^dim^ degranulating NK cells. Data represent mean +/− SD of at least 5 experiments. **p* < 0.05 relative to resting NK cells stimulated with anti-NKp46 mAb-coated beads. (**D**) Confocal microscopy observation of NK cell degranulation on CD45^+^ (green) PBMCs stimulated with anti-NKp46 plus anti-2B4. A diffuse labelling pattern. No relevant polarization of CD107a (blue) is detectable on degranulating NK cells.

**Figure 6 cells-09-00753-f006:**
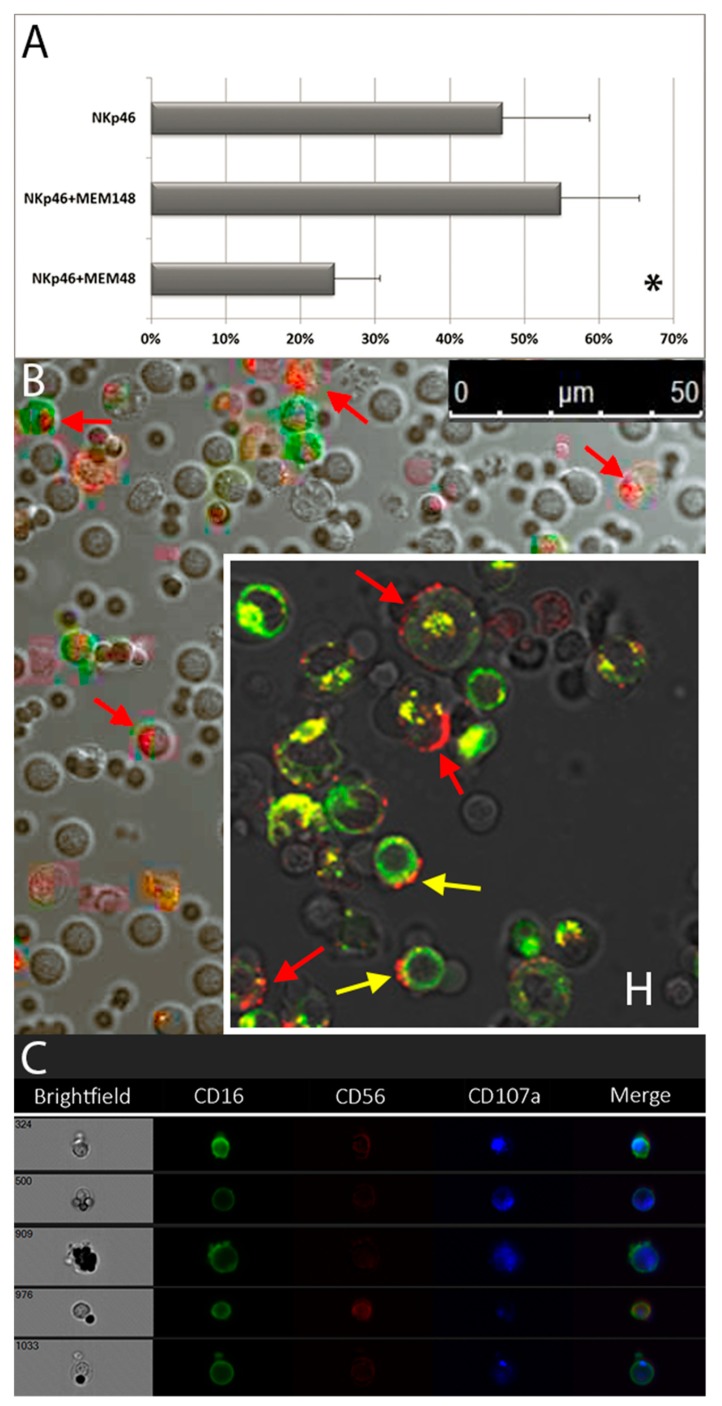
Degranulation and granule polarization analysis of in vitro IL-2-cultured NK cells after stimulation with the indicated mAb combinations. (**A**). Degranulation percentages of IL-2-cultured NK cells stimulated with anti-NKp46 with or without anti-LFA-1 β2-subunit, clone MEM-48 or MEM-148. Data represent mean +/− SD of at least 5 experiments. * *p* < 0.05 relative to NK cells stimulated with anti-NKp46 mAb-coated beads. (**B**) Confocal microscopy observation of NK granule polarization of IL-2-cultured NK cells stimulated with anti-NKp46 plus anti-2B4 and anti- LFA-1 β2-subunit (clone MEM-148) loaded microbeads or (insert H) with green-labelled Jurkat targets. Polarization of CD107a (red spots) expression are indicated by red arrows. Trogocytotic transfer of CD107a (red spots) on some Jurkat green cells (see yellow arrows) and, in some cases, of Jurkat fluorescence (green/orange/yellow spots) on some green negative lymphocytes are also evident. (**C**) ImageStream observations of NK granule polarization of IL-2-cultured NK cells stimulated with anti-NKp46 plus anti-2B4 and anti- LFA-1 β2-subunit (clone MEM-148). The polarization (spots) of CD107a (blue) is well observable.

**Table 1 cells-09-00753-t001:** Antibody specifications.

MAb	Company	Clone	Isotype	Fluorochrome
CD2 (LFA-2)	Miltenyi Biotec	LT2	IgG2b	Biotin
CD3(CD3 ε-chain)	Ancell	UCHT1	IgG1	FITC/APC
CD16(FcγRIII)	Ancell	3G8	IgG1	FITC/PE
CD18(LFA-1 β2-chain)	Immunological Sciences	MEM-48	IgG1	Biotin
CD18(LFA-1 β2-chain)	Immunological Sciences	MEM-148	IgG1	Purified *
CD45(LCA)	Ancell	C11	IgG2a, κ	FITC
CD56(NCAM)	Biolegend/Exbio	MEM-188	IgG2a	FITC/PE/Biotin
CD107a(LAMP-1)	Biolegend	H4A3	IgG1	PerCP-Cy5.5
CD158a/h(KIR2DL1/S1)	BD Biosciences	HP3E4	IgM	PE
CD158b/j(KIR2DL2/3/S2)	BD Biosciences	CH-L	IgG2b	PE
CD158e(KIR3DL1)	Miltenyi Biotec	DX9	IgG1	PE
CD159a(NKG2A)	Beckman Coulter	Z199	IgG2b	PE
CD226(DNAM-1)	AbCam	DX11	IgG1	Biotin
CD244(2B4)	eBiosciences	C1.7	IgG1, κ	Biotin
CD335(NKp46)	Miltenyi	9E2	IgG1	Biotin

* The antibody was biotinylated with the One Step Antibody Biotinylation Kit (Miltenyi Biotec).

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
