# Peer review of "Understanding the Synergy of NKp46 and Co-Activating Signals in Various NK Cell Subpopulations: Paving the Way for More Successful NK-Cell-Based Immunotherapy"

_cells, 2020, doi:10.3390/cells9030753_

Round 1

Reviewer 1 Report

In this manuscript, Loris Zamai et al. analyzed the responses of NK cells to various combinations of receptor-specific stimuli in a microbead-based stimulation system using agonistic antibodies to the natural cytotoxicity receptor NKp46, 2B4, DNAM-1, and CD2. Two aspects of the cytotoxic reaction were assessed, degranulation (in CD107a-based cytometric assay) and granule polarization (confocal microscopic and Image-stream analysis). The microbead-based stimulation is informative system, although it may not completely reproduce the NK cell activation evoked by the interaction with target cells. Authors concentrated on the analysis of more and less differentiated NK cell subsets, including licensed and unlicensed NK cells. Most significant increase in CD56dim NK cell degranulation was shown for the combination of anti-NKp46 and anti-2B4 antibodies compared to anti-NKp46 antibody alone. Topic of the article is of great interest in the context of revealing the conditions for optimal activation of NK cells required for their using in immunotherapy. Authors demonstrate the expert level in presenting scientific background of the work, performing analysis and interpretation of the results. Main conclusions are supported by experimental data. The manuscript is very well written, and read as a fascinating narrative.

Next concerns should be addressed to improve the overall quality of the manuscript.

  1. Synergistic action implies formally that the combined effect exceeds the sum of the two effects separately. To confirm the synergistic effect, the authors should provide data on degranulation when using beads with anti-2B4 only. In another case, in Results section it should be said, cooperative effect, or enhancing effect.
  2. Including in the manuscript the own data on differential surface expression of NKp46, 2B4, DNAM-1, and CD2 in the analyzed NK cell subsets is recommended.
  3. Figure 5 shows, as a control, resting NK cells stimulated with antibody-coated beads. It is necessary to provide additional data on the control LAMP-1 staining of NK cells, especially those cultured for 3 days, obtained in the same mode, since increased staining may be associated with increased permeability of the plasma membrane of dying cells. Examples of surface staining - primary data - should be given not only for ex vivo cells, but also for cultured NK cells.
  4. Throughout the Results section not only general column graphs but examples of cytometric plots should be additionally presented.
  5. The manuscript contains many indications “data not shown”. Most of these data should be presented, for example, as supplementry materials.
  6. Discussion section looks overloaded and need to be reduced. Some issues can be discussed in detail in a separate manuscript.
  7. The last statement in the subsection 4.2 (discussion) is controversial.
  8. Although the manuscript is well structured, it is too “wordy” and requires additional style refinement.
  9. Subsections 2-4 in Discussion contain a mistake in titles: B4 instead 2B4.
  10. The manuscript should be checked for misprints (lines 110, 321, etc).  

Reviewer 2 Report

This manuscript by Zamai and colleagues is a presentation of a microbead-based method for the ex vivo stimulation (assessment of function) and expansion of NK cells, with specific impactful implications for the CD56dim population (and NKp46+2B4 stimulation) in emerging cancer therapies. In general this is a straightforward manuscript.  Its contextualization with other NK cell papers is extensive – indeed this could be considered excessive by some editors, especially the discussion (100 refs total), but I personally appreciated what is essentially also a mini-review on NK cell receptors. Furthermore, such an extensive discussion of literature may be necessary as Cells is not an immunology-specific venue.  The experiments seem appropriate and well-executed and the authors have transparently stated any limitations, e.g. significant variability in response, and how they account for that. I believe this will be a valuable technical addition to the literature on NK cells. I am surprised that this was not already published in an immunology-specific journal; perhaps this is a testament to the unreasonable expectations of those venues nowadays. 

I have the following minor concerns that should be addressed prior to acceptance of this work:

  1. I suggest submitting a supplementary table that lists all of the antibodies, their clone numbers or product numbers, vendor, fluorophore, etc. While some of this information is in the methods, it would be more easily accessible in a tabular format.
  2. There are some data that the authors “do not show”. I strongly encourage submitting such data as supplementary figures/files. One area where this is especially important are the data demonstrating the dose ratios of beads-to-cells.
  3. The authors should engage with an English-language editing service. There are some major flaws in grammar and word usage throughout the manuscript (g., significant instead of significative). Additionally, I suggest being consistent with the use of “CD” vs. “LFA” terminology. It is important to state both at first instance, but it can be confusing to the reader to interchange the terminology.

Round 2

Reviewer 1 Report

The considerable revision of the manuscript was performed. The manuscript can be published in “Cells” after minor revision.

Here are my suggestions:

  1. Figure 1 shows the NK cells stimulated for 3 days with IL-2, 20 ug/ml, which can be easily divided in CD56bright and CD56dim, that is not very typical for highly activated cells. It is unclear how large the IL-2 dose was in active units. It should be indicated.
  2. There is heterogeneity in the use of “NK cells” in the manuscript. “NK cells”, “NKs” (lines 103, 286) and “NK” (lines 164, 263), “NK activated cells” (line 159) can be found. I would suggest using the universally accepted terms “NK cells”, “activated NK cells” etc. throughout the text.
  3. In Supplementary figures the quadrant percentages of cells are too small a font. They should be specified in a larger font.
